# Overload of the lower limbs of firefighters as a result of external conditions

**Dagmara Iwańska**[1]*, **Piotr Tabor**[1], **Czesław Urbanik**[1], **Ida Wiszomirska**[2], **Andrzej Mastalerz**[1]

1 Department of Biomedical Sciences, Faculty of Physical Education, Józef Piłsudski University of Physical Education in Warsaw, Warsaw, Poland, 2 Department of Physiotherapy Fundamentals, Faculty of Rehabilitation, Józef Piłsudski University of Physical Education in Warsaw, Warsaw, Poland

* dagmara.iwanska@awf.edu.pl

## Abstract

### Background

It is known that a firefighter's uniform with all equipment will significantly increase the load on the lower limbs during landing. The higher landing height will also increase ground reaction forces. However, it is unknown what effect the firefighters' experience (job seniority) will have on the load. Moreover, age-related changes in neuromuscular control, balance, and muscular strength may alter the way external loads are managed, potentially increasing susceptibility to overload.

### Objective

The present study aimed to evaluate the overload of the musculoskeletal system resulting from changing external conditions.

### Methods

The main study involved 171 firefighters divided into three age categories: up to 25 years ($I_{n=83}$), up to 44 years ($II_{n=38}$) and over 44 years ($III_{n=50}$). The subjects performed three landing on to the force platforms from two heights (0.5 m; 1.0 m) and in two types of clothes: sport and fire protection. For the evaluation of lower limb overloads, the value of generated ground reaction force relative to body weight (GRF/BW) of the participant and time of force affecting the locomotor system (amortization time) were evaluated ($t > 1$ BW).

### Results

Increasing the landing height from 0.5 m to 1 m resulted in a significant ($p < 0.001$) increase in ground reaction forces. Furthermore, younger participants generated significantly greater forces ($p < 0.001$). Performing the landing in protective clothing also

**Data availability statement:** All relevant data are publicly available from Zenodo at https://doi.org/10.5281/zenodo.17764420.

**Funding:** The work was supported by The National Centre for Research and Development* under the research grant number DOB-BIO6/05/54/2014 - Development of the sensorimotor profile and performance testing procedures for KSRG rescuers in standardized rescue activities, carried out in 2014-2018. The funders had no role in study design, data collection and analysis, decision to publish, or preparation of the manuscript.

**Competing interests:** The authors have declared that no competing interests exist.

resulted in increased load ($p < 0.001$). Furthermore, older firefighters had a longer (*$p < 0.01$*) cushioning time at both landing heights and in both clothing.

## Conclusion and application

These results suggest that younger or less experienced firefighters may employ sub-optimal landing strategies, increasing their injury risk.

---

## Introduction

Firefighters are often exposed to increased health and life risks and the risk of temporary inability to work [1,2]. This is due to the specific and complex load associated with physical exercise characterised by different types of muscle work and load, and different levels of exercise intensity. Therefore, it is the musculoskeletal system that is exposed to injuries and sprains most often [1,3–5]. Dealing with one's own weight as well as with external load may lead to substantial joint overload, e.g., when walking and running [6]. In the case of rescue operations, a multifold increase in the load to the musculoskeletal system can be noted, which may result in damage to both passive and active parts of the system [7].

Moreover, occupational demands in firefighting persist across a wide age span, from younger recruits to experienced older personnel. Age-related changes in neuromuscular control, balance, and muscular strength may alter the way external loads are managed, potentially increasing susceptibility to overload. Despite this, limited research has examined how age interacts with physical demands typical of firefighting tasks, such as impact landings under external load.

When walking, ground reaction forces are transferred to human bones, muscles and joints, thus reducing the strain on the musculoskeletal system. However, these forces are considerable when the joints of the supporting limb are locked as is the case with drop jumps from substantial height [8]. The factors that influence ground reaction forces also include foot impact speed, foot angles in the sagittal and frontal planes at the moment of ground contact, energy absorption by the muscular system [9] the shoe sole type, pattern and stiffness, as well as the type of surface the person jumps on [10]. Despite existing knowledge about factors influencing ground reaction forces (GRF), few studies have assessed how these variables behave under typical firefighting conditions, such as jumps from fire vehicles or movements while wearing heavy gear.

All these factors are exacerbated by the height of the jump. In case of fire in a tall building, firefighters evacuate people from fire-affected floors using firefighting ladders or aerial platforms. When leaving the cabin of a fire engine, his body weight also acts vertically downwards. In fire engines, the driver's cabins are usually one meter above the level of the ground, whereas the height of the first step is about 50 cm. Giguere and Marchand [8] noted that of all accidents recorded during the firefighting operations, 37% of the injuries occurred when leaving the car. Giguere and Marchand [8] conducted a study to document ground reaction forces acting on firefighters

wearing personal protective equipment when they descended a fire engine in various ways (forward or backward to the fire engine's cabin). These researchers revealed significantly lower reaction forces (exceeding 3.2 times the body weight) when getting off the fire engine forward than backward. These findings highlight the importance of proper movement technique to minimize biomechanical load, particularly during landings. For this reason, an essential factor in reducing stress while jumping is the technique of performing the movement itself, which is associated with impact absorption during the landing [11]. The increase in impact absorption during jumps is related to the skill acquired during many years of training and may be related to the proportion of the strength of the agonist and antagonist muscles of the knee joint as well as the work of the muscles in the stretching-contraction cycle. The hamstring-to-quadriceps strength ratio (H:Q) is considered an important parameter in this context, as it affects knee joint stability and the ability to absorb vertical forces during landing. This ratio is considered as an indicator for injury prevention, evaluation of rehabilitation outcomes, and is a criterion for returning to activity after an injury [12]. However, previous studies have not clearly linked H:Q ratios with GRF changes in firefighters under different external load conditions. In all the cases described above, the necessity to wear special gastight bunker gear and a breathing apparatus constitutes a huge burden for rescuers. Full firefighter's protective clothing consists of clothing (3.5 kg), a helmet (1.5 kg), shoes and a breathing apparatus (13 kg). This gear effectively protects the rescuer from environmental hazards; however, extra weight and visibility restrictions increase energy expenditure [13–16], reduce freedom of movement and musculoskeletal system efficiency, and can increase load to the musculoskeletal system when moving, jumping, jumping off and dealing with external loads [17–19].

Protective gear worn during rescue operations introduces an additional load and restricts natural body movement. While it is known that such equipment increases energy expenditure, it remains unclear how it specifically affects landing dynamics and joint loading. Understanding the effect of gear on lower limb overload is essential for injury prevention and may inform improvements in equipment design or training protocols. Given these knowledge gaps, there is a strong need to investigate how both external constraints (such as firefighting gear) and intrinsic factors (such as age-related neuromuscular adaptation) influence the ability to absorb landing forces. A better understanding of these interactions may lead to more effective training, injury prevention strategies, and operational safety guidelines tailored to age and equipment use. Ttherefore, taking into account the combined biomechanical burden of external equipment and dynamic activities such as jumping, there is a justified need to assess whether different firefighter populations – especially those differentiated by age – experience varying levels of musculoskeletal overload. Additionally, it is relevant to explore whether neuromuscular parameters such as the hamstring-to-quadriceps torque ratio may serve a compensatory role in force absorption. The present study aimed to evaluate the overload to the musculoskeletal system resulting from changes in external conditions. Specifically, it examined how the use of protective clothing and differences in age affect ground reaction forces during jumps from different heights. The paper attempts to find out to what extent the specificity of the firefighting work and experience affects the ability to reduce strain to which the musculoskeletal system is exposed when jumping from different heights and to what extent does the flexor/extensor muscle torques ratio affect impact absorption in lower limbs. By identifying the relationship between H:Q ratio and landing biomechanics, this study also seeks to assess whether muscular balance may serve as a compensatory factor that reduces injury risk in dynamic tasks under load.

## Material and methodology

### Research material

The study included 179 male firefighters from multiple Polish fire stations. The subjects were recruited from 01.04.2015 to 31.07.2015. Participation in the study was voluntary. All of the firefighters underwent a qualification procedure which involved filling in a questionnaire on the health status of the respondent, a cardiological and orthopedic examination.

The inclusion criteria were as follows: providing informed, written consent to participate in the research, age of between 20 and 50 and actively working as a firefighter.

The exclusion criteria included: medical contraindications (cardiovascular diseases, chronic diseases such as diabetes, asthma, lung disease), and orthopaedic problems (injuries: lower limbs, spine) occurring within less than 0.5 years.

A medical examination was performed by a medical doctor and an orthopedist. Details about the study procedures of the recruitment, qualification phase and main study are provided in the timing diagram (Fig 1).

Of the 179 firefighters, 171 qualified for the main study. Six people did not qualify for the main study due to exclusion by the medical doctor. This was due to cardiac problems and problems related to chronic diseases (diabetes). The subjects were divided into three age groups: 20–30 years old ($I_{n=83}$), 30–40 years old ($II_{n=38}$) and 40–50 years old ($III_{n=50}$). The criteria for selecting participants for age category were related to minimum, average and long experience of active work. The participants who were included in the study were informed about the purpose and the method of preparation for the measurements. The subjects consented to the research.

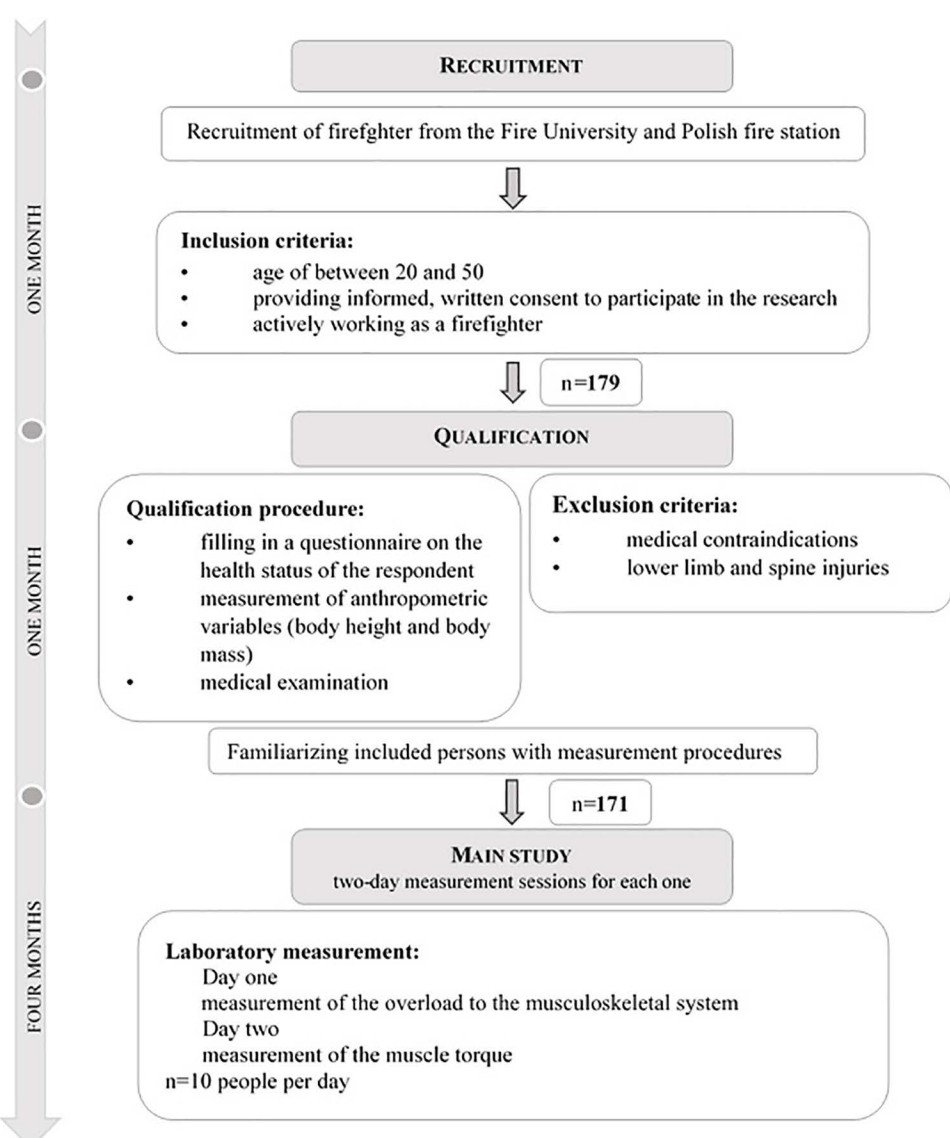

**Fig 1. The timeline and consort diagram of study procedures of the recruitment, qualification phase and main study.**

The consent of the ethics committee was obtained to conduct the research (Ethics Committee for Human Research at the Military Institute of Aviation Medicine in Warsaw; nr 01/15).

## Data analysis

The measurements of basic anthropometric parameters were performed, i.e., body height (bh), measured with an accuracy of 0.5 cm using an anthropometer, and body mass (bm) measured with an accuracy of 0.5 kg.

The evaluation of the overload to the musculoskeletal system was carried out for drop jumps in different external conditions. The participants performed three backward drop jumps onto two piezoelectric Kistler platforms at 5-second intervals, with each lower limb landing on a separate platform. These tests were performed from two heights (0.5 and 1 m) and in two different outfits (sports and fire protection clothes). The bunker gear increased the load by 75 N on average. Based on three jumps, one averaged profile of vertical ground reaction forces was developed for each test. Averaging over time was performed with maintaining the total impulse of the force generated during the test (1)

$$\int_{t_0}^{t_k} F_1 dt = \int_{t_0}^{t_{av}} F_1' \ dt$$

(1)

where: $t_0$-$t_k$ – test duration; $t_0$-$t_{av}$ – averaged time; $F_1$ – vertical ground reaction forces; $F_1'$ – forces after transformation.

For the evaluation of lower limb overload, the value of generated ground reaction forces (GRF) relative to body weight (BW) of the participant and time of force affecting the musculoskeletal system ($t_{abs}$ – impact absorption time) were used (t > 1 BW). Fig 2 shows an example diagram of vertical ground reaction forces recorded during one landing.

The respondents performed the tests in the following order: platform height of 0.5 m and 1 m, in sportswear which consisted of a T-shirt, short comfortable shorts and soft sports shoes and after using the same order (0.5 and 1 m), the tests were repeated in firefighter's protective clothing which consisted of a special suit, gloves, a helmet and stiff fireproof footwear (without the air cylinder and a gas mask).

The muscle torque (T) of knee jonts extensors (ext) and flexors (flex) were measured (Fig 3). The measurement was performed under static conditions in a sitting position and in the 90° position in the knee joint according to the procedure proposed by Knapik et al. [19]. The angle value used in the tests stemmed from the characteristics of static muscles, based on which the conditions needed for producing maximum torque are determined [20].

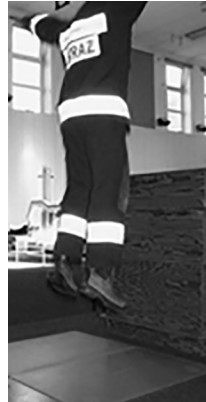 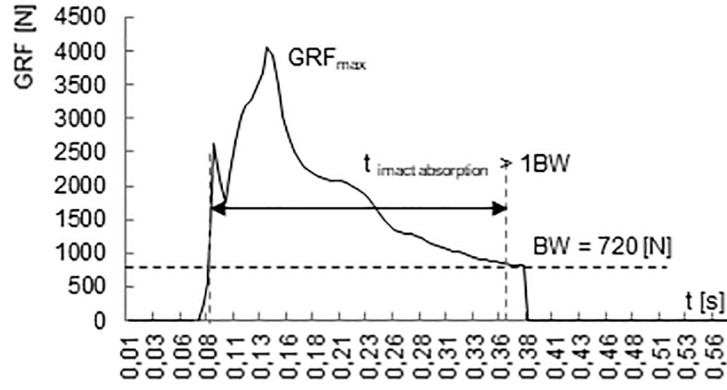

**Fig 2. Example of ground reaction forces during a drop jump from the height of 0.5 m in sportswear expressed in absolute units together with a graphical presentation of the parameter determination methodology.**

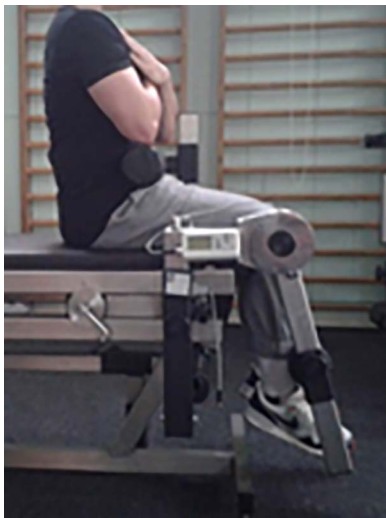

**Fig 3. The measurement position for torque of the knee extensors in which the static conditions of the bone lever (lower leg) were preserved.**

The subjects performed two isometric contractions lasting 3–5 s with a rest break of 15 s and maximum values were included in the analysis. Prior to the test, each participant did a 10-minute warm-up.

Joint stability evaluation as the flexor/extensor muscle torques ratio was calculated using the formula:

$$H/Q ratio = (1 - (T_{flex}/T_{ext})) \times 100\%$$

(2)

were: $T_{flex}$ – value obtained for flexors; $T_{ext}$ – value obtained for extensors [21]. Therefore, the calculated index of proportion of muscle strength for the knee joint was analysed. Results above 50% point to a twofold (or greater) advantage of lower limb extensors, whereas values below 50% indicate an increasing participation of flexors.

## Statistical analysis

The recorded data were analysed statistically using the STATISTICA software (v.12). We chose appropriate methods of statistical analysis which took into consideration all external factors leading to the musculoskeletal system overload (jumping height, repeatability, clothing, and age of the participants). For this reason, the analysis of variance ANOVA with repeated measures design was used. The effect size was assessed based on the partial η2 value (0.01–0.06: small effect, 0.06–0.14: medium effect, > 0.14: large effect).

If interactions between the factors were found to have a significant effect, repeated measures test (Tukey *post hoc* test) was employed in further analysis. Mauchly's sphericity test was applied to check if its assumption was met for the tests used. If it was violated, Greenhouse-Geisser (G-G) correction was applied. As the data were normally distributed, the interrelationships between of variables characterizing vertical overload of the lower limbs (GRF/BW and impact absorption time) were examined by Pearson's simple correlation coefficients. For all tests, the probability level limits were set at $p < 0.05$.

## Results

The mean ±SD anthropometric variables describing particular groups are presented in Table 1.

The mean body height for all groups were similar. The differences between groups did not exceed 1.5% and were not statistically significant ($F_{(2,171)}$ = 2.8; $p < 0.062$; $η^2 = 0.036$). The lowest mean body weight was found in group I, while the

**Table 1. Mean ±SD of age years, and anthropometric variables: body height (bh cm) and body mass (bm kg), and range of variation describing particular groups.**

| age groups | I $_{n=83}$ | II $_{n=38}$ | III $_{n=50}$ | F $_{df=2.168}$ | p | η² |
|---|---|---|---|---|---|---|
| age years | 22.2 ±1.91 | 36.2 ±2.73 | 44.2 ±3.14 | | | |
| bh cm | 180.6 ±6.80 | 179.9 ±4.79 | 178.0 ±5.03 | 2.8 | <0.062 | 0.036 |
| bw kg | 79.5 ±8.36 | 89.7 ±9.39 | 87.2 ±13.25 | 16.5 | <0.001 | 0.164 |
| BMI | 24.3 ±1.85 | 27.7 ±2.64 | 27.5 ±3.52 | 32.7 | <0.001 | 0.281 |

highest was found in the second and the oldest group. For this variable, a significant age effect was found ($F_{(2,168)}$ = 16.5; $p<0.001$; $η^2=0.164$). The differences between the first group and the remaining groups were 12.8 and 9.8% and were significant ($p<0.001$). Even greater statistically significant differences were found for BMI (13.8 and 12.9%). For this variable, the significant effect of age was ($F_{(2,168)}$ = 32.7; $p<0.001$; $η^2=0.281$).

The results showed that for each age group and each height of the drop jump performed either in sportswear or bunker gear, there was an increase in ground reaction forces generated during jumps. However, statistical analysis failed to demonstrate interactions between these factors (clothing, jumping height and age) for both variables. Means with standard deviation for the results of the vertical overload (GRF/BW) are shown on Fig 4 and the impact absorption time ($t_{abs}$) shown in Table 2.

Effect for:

Age – $F_{(2,170)}$ = 9.11; $p < 0.001$***; $η^2$= 0.098.

Type of clothes – $F_{(2,170)}$ = 36.47; $p < 0.001$***; $η^2$ = 0.178.

Height– $F_{(2,170)}$ = 450.57; $p < 0.001$***; $η^2$ = 0.728.

The effect of age, clothing and height was observed for GRF/BW (Fig 4). It was concluded that the value of the produced ground reaction forces exceeded body weight fivefold and significantly changed with the participants' age. This also concerned the time when the load affected the musculoskeletal system (Table 2). The men from the oldest group manifested the lowest load during drop jumps from both 0.5 and 1 m compared to the younger groups. The height of the

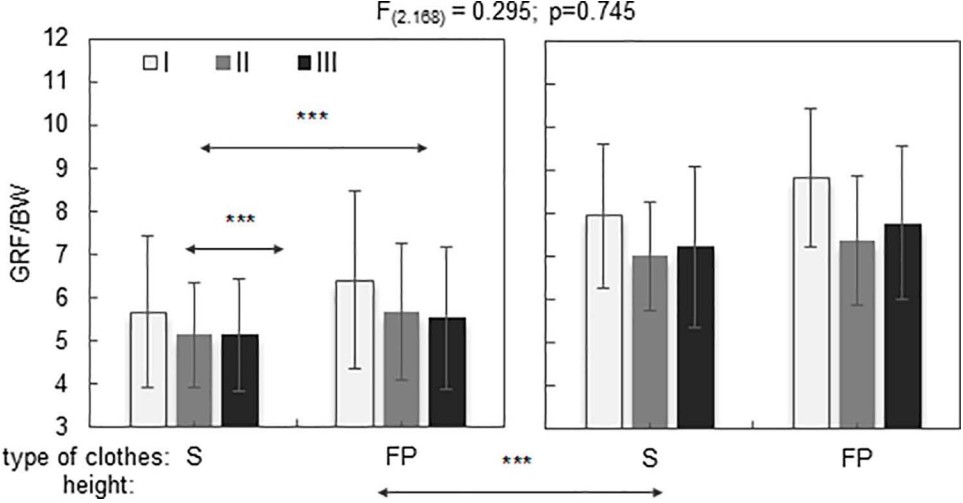

**Fig 4. Mean values±SD of the vertical overload (GRF/BW) during drop jumps from different heights (0.5; 1 m) and in different clothing (S-sport, FP-fire protection) for different age groups (I; II; III).**

**Table 2. Mean values ±SD of the impact absorption time (t$_{abs}$) during drop jumps from different heights (0.5; 1 m) and in different clothing (S-sport, FP-fire protection) for different age groups.**

| age groups | I n=83 | | II n=38 | | III n=50 | |
|---|---|---|---|---|---|---|
| type of clothes | S | FP | S | FP | S | FP |
| height | impact absorption t s > 1BW | | | | | |
| 0.5 m | 0.29±0.10 | 0.36±0.14 | 0.36±0.12 | 0.38±0.14 | 0.38±0.12 | 0.40±0.13 |
| 1 m | 0.41±0.14 | 0.48±0.12 | 0.46±0.09 | 0.47±0.14 | 0.47±0.11 | 0.50±0.13 |

Effect for:

Interaction: $F_{(2,168)}$ = 0.142; $p = 0.867$.

Age: $F_{(2,168)}$ = 5.862; $p < 0.01$; $\eta^2 = 0.065$.

Clothes: $F_{(2,168)}$ = 21.44; $p < 0.001$; $\eta^2 = 0.113$.

Height: $F_{(2,168)}$ = 140.14; $p < 0.001$; $\eta^2 = 0.455$.

landing has a significant influence on the values higher reaction forces, however the effect of this factor was similar in all age groups. The average differences in GRF/BW between 0.5 m and 1 m tests were significant (group I – 39%, group II – 33%, group III – 40%).

The analysis of the second variable related to the time of the force acting during the landing, i.e., impact absorption time, revealed (also in this case) that the oldest firefighters demonstrated the greatest ability to absorb overload, as evidenced by significantly longer impact absorption. Also it was shown that impact absorption time was significantly influenced by individual factors (age, clothing) and significant clothing/age interaction, but the highest extent, jump height.

A significantly lower level of the ability to absorb the force acting on the musculoskeletal system was found in the youngest group. The participants from group I manifested higher values of GRF/BW and had significantly shorter impact absorption time compared to the older groups ($p < 0.001$). However, attention should be paid to the extended load impact absorption time during drop jumps performed by the youngest firefighters wearing fire protection clothes.

Table 3 shows the result of the analysis of the correlation between the vertical overload (GRF/BW) and the impact absorption time (t$_{abs}$), which was performed for all participants.

The values of Pearson's simple correlation coefficients showed statistically significant negative relationships between of the vertical overload (GRF/BW) and impact absorption time in almost for all trials in second and oldest groups. Such a relationship was not demonstrated during a drop jump from a height of 1 m in a fire protection clothes. In the youngest group, no statistically significant correlation was found. The results of the significant relationship were presented in Fig 5.

**Table 3. Values of Pearson's simple correlation coefficients and probability levels of variables characterizing vertical overload (GRF/BW) and impact absorption time (t$_{abs}$).**

| | | GRF/BW | | | |
|---|---|---|---|---|---|
| drop jump height | | 0.5 m | | 1 m | |
| type of clothes | | S | FP | S | FP |
| t$_{abs}$ [s] | I | 0.235 $p=0.883$ | 0.049 $p=0.660$ | −0.072 $p=0.514$ | 0.106 $p=0.341$ |
| | II | −0.207 $p=0.212$ | −0.344 $p<0.05$ | −0.412 $p<0.01$ | −0.077 $p=0.647$ |
| | III | −0.429 $p<0.01$ | −0.558 $p<0.001$ | −0.427 $p<0.01$ | −0.201 $p=0.093$ |

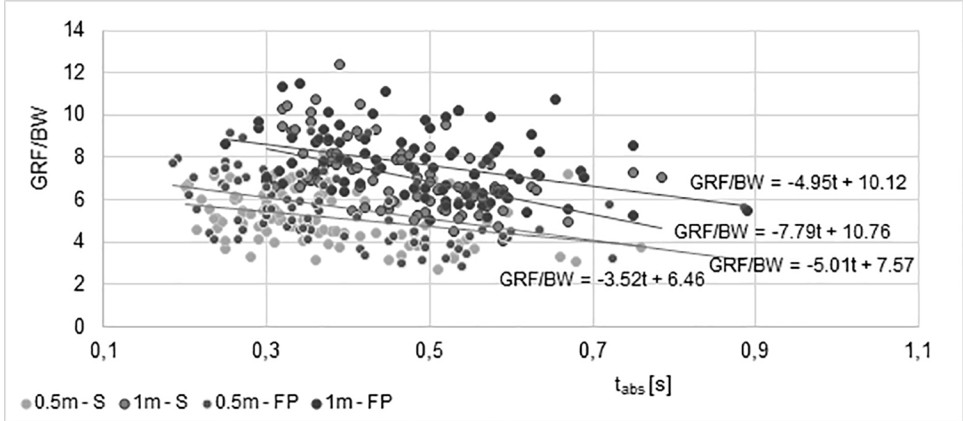

**Fig 5. Changes in GRF/BW values as a function of impact absorption time ($t_{abs}$) approximated by the linear regression equation for each performed height trial (0.5 and 1 m) and different attire (S-sport; FP-fire protection).**

The GRF/BW results obtained in groups II and III were approximated by the linear regression equation. The coefficient b of the equation indicates how quickly the overload (GRF/BW) will decrease if the impact absorption time is extended. The decrease in GRF/BW values as a function of impact absorption time is presented separately for each performed trial.

Further analysis concerned the evaluation of muscle strength and H/Q ratio of the lower biokinematic chain. Mean ±SD values for the sum of muscle torques for knee extensors and flexors are shown in Table 4.

A significant age effect on ST values was noted ($F_{(2,168)}$ = 9.99; $p < 0.001$). The youngest and the second group demonstrated significantly higher ST values compared to the group of oldest firefighters. The difference was found between groups I and III (12.7%) as well as II and III (9.3%).

However, the main goal of the measurements was to calculate the H/Q ratio (the stability index for the knee joint). The proper proportion between antagonistic and agonistic muscles ensures adequate muscle functioning as well as stability of a joints. It was the flexor/extensor muscle torques ratio (H/Q ratio) was similar in all age groups. Based on the analysis of variance, we cannot talk about any significant changes in the age function ($F_{(2,168)}$ = 0.634; $p = 0.532$).

The H/Q ratio was subjected to correlation analysis with the results illustrating lower limb overload (GRF/BW). A significant correlation was found in the youngest group of firefighters ($p < 0.05$) in two jump trials from 0.5 m in both sports and firefighting attire (Fig 6).

It was observed that firefighters with weaker extensor muscles relative to flexor muscles also exhibited higher lower limb overload values. These correlations, the same as above, were presented using the linear regression equation. However, such correlations were not observed during jumps from 1 m. Additionally, in the other groups of firefighters, the proportion index did not correlate with the GRF/BW results.

**Table 4. Mean ±SD values for the sum of lower limb muscle torques (ST) and the hamstring/quadriceps ratio (H/Q ratio) obtained in particular age groups (I – up to 25 years, II – 25 to 44 years, III – over 44 years).**

| variables | age groups | I $_{n=83}$ | II $_{n=38}$ | III $_{n=50}$ | F $_{df=(2.168)}$ | p | η² |
|---|---|---|---|---|---|---|---|
| **ST** | Mean±SD | 385.2±62.2 | 371.1±63.1 | 336.4±57.7 | 9.99 | *<0.001* | 0.106 |
| **H/Q ratio** | | 42.8±10.0 | 40.3±13.8 | 41.8±10.9 | 0.634 | *=0.532* | 0.008 |

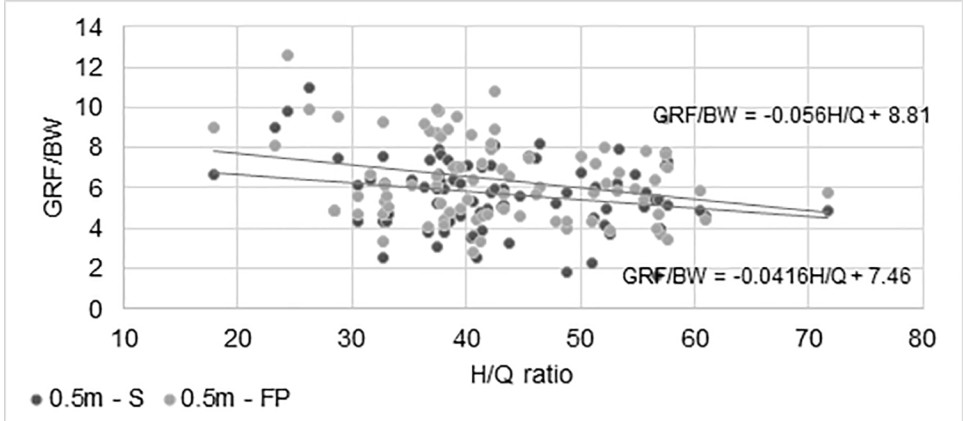

**Fig 6. Changes in GRF/BW values as a function of H/Q ratio approximated by the linear regression equation for each height trial (0.5 and 1 m) and different attire (S-sport; FP-fire protection).**

## Discussion

The overload to which the firefighters' musculoskeletal system is subjected during rescue operations results from the necessity to work in unknown, constantly changing external conditions often accompanied by low visibility. During each operation, a firefighter performs activities that include walking, jumping, landing after jumping off, overcoming obstacles, climbing stairs, jumping down stairs and getting off vehicles [8,22–24].

For this reason, the present study attempted to evaluate the ability to absorb the impact of landing during drop jumps from different heights in firefighters of different ages and on different levels of external load. Firefighters' experience (age) and different types of clothing (sport and fire protection) were taken into consideration for the participants performing jumps from two heights (0.5 and 1 m). Ground reaction forces produced upon landing were analysed and compared, whereas overload (expressed as a multiple of body weight) and impact absorption time were evaluated. It was assumed that the overload to the musculoskeletal system depended on overcoming one's own weight as well as external load, i.e., firefighter's protective clothing, which consisted of a special suit, gloves, a helmet and stiff fireproof footwear. Previous studies have shown that such gear obviously increases a firefighter's safety but at the same time, it limits his or her mobility and causes earlier fatigue and reduced physical fitness, which leads to increased injury risk [25,26].

The weight of the special gear in which the drop jumps were performed in this study was 75 N on average. It seems that eventually this value increased the participants' weight insignificantly (by 8%). However, it was revealed that for the drop jumps performed in protective clothing from both heights (0.5 and 1 m), peak ground reaction forces were higher in each age group compared to the jumps performed in sportswear. Heineman et al. [3] and Park et al. [24] suggest that increased mass can only partially explain an increased incidence of injuries caused by falls (46.7%) among firefighters.

The current study found that firefighters in all groups manifested the similar stability of the knee joint. The hamstring/quadriceps ratio (H/Q ratio) ranged from 43% (I group), 40% (II group) to 42% in the oldest group. The difference between the groups was not significant. These results, while within a range observed in some studies on physically active individuals, are on the lower end of the desirable spectrum. Typically, the H/Q ratio is expected to range between 45–60%, with approximately 50% being considered optimal for healthy knee joint function, depending on the measurement method, angular velocity, and joint [21,27,28]. Values below 45% may indicate insufficient hamstring strength relative to quadriceps, which could compromise joint stability and increase the risk of anterior cruciate ligament (ACL) injuries and other knee pathologies.

In our study, joint stability evaluation as the flexor/extensor muscle torques ratio was measured under static conditions in the 90° position in the knee joint, based on which the conditions needed for producing maximum torque are determined [20]. Although isokinetic assessments are more commonly used (often at angular velocities of 60–180°/s), static measurements at 90° are also valid and are commonly used in functional assessments and ergonomics. When interpreting such data, it is crucial to consider the measurement modality.

The values obtained (40–43%) are slightly below the recommended threshold and could suggest a relative quadriceps dominance, potentially affecting knee stabilization during dynamic activities such as landing from height or sudden direction changes. However, this result is not unusual in firefighter populations. For example, Miratsky et al. [29] reported similar H/Q ratios in professional firefighters, ranging from 42 to 45%. This may reflect occupational adaptations, particularly due to repetitive load-bearing tasks involving prolonged squatting, stair climbing, or carrying equipment, which can lead to a disproportionate strengthening of the quadriceps relative to the hamstrings.

Furthermore, studies indicate that both excessively low (<40%) and high (>60%) H/Q ratios may be associated with elevated injury risk due to muscular imbalance—either from quadriceps dominance or from overactive hamstrings with weak quadriceps [28,30]. In this context, although the measured H/Q ratios in our study are not critically low, they should still be interpreted with caution, and preventive strength training focused on hamstring development may be advisable for this population.

However, significant correlations between the H/Q ratio and GRF/BW were demonstrated only among the youngest firefighters in two trials. Such correlation was not been observed in groups II and III, which may confirm that the ability to reduce lower limb loading when landing in the older groups is more closely related to a non-specific landing technique resulting from the stiffness of the footwear.

Such significant values of the ground reaction forces produced may also be linked to a decrease in impact absorption capacity during the landing in stiff footwear that changed movement kinematics in the study participants. This observation is consistent with the findings of the study conducted by Vu et al. [11] The authors examined the effect of footwear on the biomechanics of the firefighters' lumbar spine during jumps from different heights and found that the decrease in mobility in the talocrural joint significantly increased the overload to the lumbar spine. This is also due to the fact that during the landing, forces from distal parts of the biokinematic chain are transferred to its proximal parts or dispersed [31,32]. Therefore, the lowest joint has the greatest impact on the kinematics of the entire chain and impact absorption on landing.

However, proper footwear design is not the only factor reducing the overload to the musculoskeletal system. The findings of the present study showed that the drop jump height was the greatest determinant of the overload to the firefighters' musculoskeletal system in all the compared age groups. This factor produced statistically stronger effects when comparing the influence of the increased weight caused by wearing protective clothing. The disproportion between the external load resulting from the drop jump height was probably linked to the strength of the musculoskeletal system which, during the jump from the height of 1 m, could not keep up with increasing mechanical stresses during the impact absorption phase.

Melińska et al. [33]. showed that the value of the ground reaction forces recorded during drop jumps rises together with an increase in the step height and, compared to the value of the force while standing, grows three to four times. Lees [34] performed measurements of ground reaction forces during the landing phase from greater heights (1.0 m) and emphasised impact absorption time. The author demonstrated that although the entire landing phase took more than 1.0 s, impact absorption lasted only 0.15–0.2 s, while the rest of the landing phase involved maintaining balance. This was also confirmed by the recorded levels of ground reaction forces, which differed in the further part of the landing phase. Therefore, the author divided the landing phase into hard landing and soft landing. During hard landing, the body adopted a more vertical position; small flexion was recorded in the main joints of the lower limbs, whereas impact forces were greater but they lasted for a relatively shorter period of time. Such reactions were observed in the present study in the youngest firefighters. Regardless of the height and the clothing type in which they performed the drop jumps, the impact

of reaction forces on the musculoskeletal system was significantly greater, with significantly shorter impact absorption time than in experienced firefighters (older age groups). The landing in this group was considered to be hard. The values of reaction forces produced in this group exceeded 5.6BW when landing in sportswear from the height of 0.5 m and reached as much as 8.9BW in bunker gear when jumping from the height of 1.0 m. Furthermore, impact absorption time ranged from 0.28 to 0.48 s. According to Lees [34], the main joint involved in hard landing impact absorption is the knee joint, whose flection was intended to slow down the movement of the body trunk. For soft landing, characterised by longer duration and lower values of the recorded force, the muscles were gradually activated in subsequent body segments, leading to a slowdown in the movement of the entire body. It can be assumed that more experienced firefighters intuitively slowed down the movement of the entire body, and consequently, the overload to the musculoskeletal system was significantly smaller than in the group of the youngest participants. It is important to damp the forces during drop jumps to relieve the musculoskeletal system. This is consistent with the study by Mizrahi and Susak [35,36]. Using a dynamometric platform, these authors recorded ground reaction forces after a free fall from two heights similar to our study (0.5 m and 1.0 m). At the same time, they filmed movement and used the video recordings to determine angles in the lower limb joints and accelerations of selected points located on the body. Based on the obtained results, it was found that an increase in hip joint flexion improved impact absorption upon landing and decreased the values of the recorded reaction forces. The researchers also stressed that landing on toes resulted in lower reaction forces and longer impact absorption time, as opposed to heel landing, where the recorded forces reached much higher levels, while impact absorption time was much shorter.

The heights from which the firefighters jumped were imposed for some reasons. This is due to the height at which the steps in fire engines are mounted (0.5 m) and the height of the crew cabin (1.0 m). Injuries related to rescue vehicles account for 19% of the accidents that occur during firefighting operations, 37% of which take place while getting off the fire engine [5]. Giguere and Marchand [8] compared the values of ground reaction forces produced during the descent from five different locations on fire engines. These authors revealed that reaction forces can be reduced by fixing various grips and rails for better motion control and lower falling speed. The authors also suggested the use of a proper technique for getting out of the fire engine facing the cabin. Respondents who faced the street developed much greater reaction forces on landing. On average, they were 3.2 times higher than the body weight of the participant. This method of getting off the fire engine was perceived by firefighters as the most dangerous. The authors also suggested that when redesigning the access to rescue vehicles, both safety needs and the reduction in biomechanical load on firefighters should be taken into account.

## Conclusion and application

A limitation of our study was the lack of randomization in the recruitment of study participants. However this study demonstrates that the magnitude of musculoskeletal overload in firefighters during drop jumps is significantly influenced by both the height of the jump and the type of protective clothing worn. The most substantial impact on ground reaction forces was observed when jumps were performed from a height of 1.0 meter in bunker gear, with forces reaching up to 8.9 times body weight. These findings highlight that jump height is a more critical determinant of mechanical load than the additional weight from protective gear alone.

The youngest group of firefighters exhibited the highest ground reaction forces, shortest impact absorption times, and reduced knee joint stability compared to older colleagues. These results suggest that younger or less experienced firefighters may employ suboptimal landing strategies, increasing their injury risk. The limited mobility caused by rigid firefighting footwear likely exacerbates this problem by impeding effective force attenuation mechanisms at the ankle joint.

Although no significant differences in the hamstring/quadriceps (H/Q) ratio were found between age groups, all participants demonstrated values below the normative range, which may indicate a generally elevated risk of knee joint instability within this occupational group.

Taken together, the findings support the need to incorporate landing technique training into firefighter physical preparation programs. Instruction should emphasize biomechanical adaptations such as increased knee and hip flexion during landings, particularly when wearing stiff protective footwear that restricts ankle movement. Additionally, optimizing firefighting footwear to allow for greater ankle mobility — without compromising safety — may reduce impact forces and improve movement efficiency.

Future research should focus on evaluating the long-term effects of targeted landing training and ergonomic interventions, including footwear redesign, on injury prevention and performance among firefighters.

---

## Key point

1. Young firefighters have greater strength capabilities in the lower limbs, but also generate a greater force impulse when landing after jumping.

2. Young firefighters are more susceptible to lower limb strain and injuries.

3. The habit of landing correctly should be learned.

---

## Author contributions

**Conceptualization:** Dagmara Iwańska, Piotr Tabor, Czesław Urbanik, Ida Wiszomirska, Andrzej Mastalerz.

**Data curation:** Dagmara Iwańska, Piotr Tabor.

**Formal analysis:** Dagmara Iwańska.

**Funding acquisition:** Czesław Urbanik, Ida Wiszomirska, Andrzej Mastalerz.

**Investigation:** Dagmara Iwańska, Piotr Tabor, Ida Wiszomirska.

**Methodology:** Dagmara Iwańska, Piotr Tabor, Czesław Urbanik, Ida Wiszomirska, Andrzej Mastalerz.

**Resources:** Dagmara Iwańska, Piotr Tabor, Czesław Urbanik, Ida Wiszomirska, Andrzej Mastalerz.

**Writing – original draft:** Dagmara Iwańska, Piotr Tabor.

**Writing – review & editing:** Dagmara Iwańska, Piotr Tabor, Czesław Urbanik, Ida Wiszomirska, Andrzej Mastalerz.

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
