## [Decision Letter · Decision Letter 0]

20 Jun 2025

PLOS ONE

Dear Dr. Iwańska,

Thank you for submitting your manuscript to PLOS ONE. After careful consideration, we feel that it has merit but does not fully meet PLOS ONE’s publication criteria as it currently stands. Therefore, we invite you to submit a revised version of the manuscript that addresses the points raised during the review process.

We look forward to receiving your revised manuscript.

Kind regards,

Shahnawaz Anwer, PhD

Academic Editor

PLOS ONE

Journal Requirements:

The work was supported by the NCBiR No. DOB-BIO6/05/54/

4. Please expand the acronym “NCBiR” (as indicated in your financial disclosure) so that it states the name of your funders in full.

The work was supported by the NCBiR No. DOB-BIO6/05/54/

The work was supported by the NCBiR No. DOB-BIO6/05/54/

7. When completing the data availability statement of the submission form, you indicated that you will make your data available on acceptance. We strongly recommend all authors decide on a data sharing plan before acceptance, as the process can be lengthy and hold up publication timelines. Please note that, though access restrictions are acceptable now, your entire data will need to be made freely accessible if your manuscript is accepted for publication. This policy applies to all data except where public deposition would breach compliance with the protocol approved by your research ethics board. If you are unable to adhere to our open data policy, please kindly revise your statement to explain your reasoning and we will seek the editor's input on an exemption. Please be assured that, once you have provided your new statement, the assessment of your exemption will not hold up the peer review process.

8. Please amend your list of authors on the manuscript to ensure that each author is linked to an affiliation. Authors’ affiliations should reflect the institution where the work was done (if authors moved subsequently, you can also list the new affiliation stating “current affiliation:….” as necessary).

9. Please include a copy of Table 34 which you refer to in your text on page 13.

10. We note you have included a table to which you do not refer in the text of your manuscript. Please ensure that you refer to Table 4 in your text; if accepted, production will need this reference to link the reader to the Table.

Reviewers' comments:

Reviewer's Responses to Questions

**Comments to the Author**

1. Is the manuscript technically sound, and do the data support the conclusions?

Reviewer #1: Yes

Reviewer #2: Partly

2. Has the statistical analysis been performed appropriately and rigorously?

Reviewer #1: Yes

Reviewer #2: Yes

3. Have the authors made all data underlying the findings in their manuscript fully available?

Reviewer #1: Yes

Reviewer #2: No

4. Is the manuscript presented in an intelligible fashion and written in standard English?

Reviewer #1: Yes

Reviewer #2: Yes

Reviewer #1: 1-The age distribution of participants in the abstract and the text of the article is different and not the same.

2- I suggest that a schematic image or time line of the study protocol and the jumps be included in the text of the article.

3. There is no discussion anywhere about the shoes of the participants. Are the shoes different in athletic and firefighter uniforms?

4. Did anyone drop out of your study? What are the exclusion criteria?

Reviewer #2: The authors have assessed three age groups pf firefighters with no support for age groups or age effect in the introduction. I can’t understand why three age groups of firefighters are included. Please clarify.

Abstract

The structure of abstract does not follow a standard manner. This part must be rewritten.

Background, it is not logical that you completed this part with a question. Better to provide a background regarding to firefighters and their fire protection equipment which leads to overload and then your question.

Results, please provide your findings objectively with p-values.

Introduction

The flow of idea needs to be strengthened. After reading this section for several times, I cant understand what is the novelty and the rationale. This is simple to say that the authors have investigated the effect of two levels of clothes and three age groups on GRF. Please provide support for your outcomes. Moreover, I did not find any support for knee joint stability (H:Q) through the introduction. In this case, I could think the authors have collected any available variable with no rationale. Please rewrite the introduction accordingly.

Methods

Oreder of conditions is not randomized. Please acknowledge it in the studylimitations.

L130-137 It must not be appeared here.

L215 and height

Discussion

L229 is unknown

L303-308 remove

L318-326 please discuss about your results, not review the literature

L335-336 different types of evaluating H/Q are existing. Mostly are isokinetic assessments. Here, based on your values and existing literature, please judge whether the values are low or not.

The reference Mbada et al., 2021 is not in the references list.

The reference Tabor et al., 20221 is not valid for your claim.

L341 please add reference

L352 Vy et al is not in the references list

L360-361 you have not plantar flexion data. Please avoid leaps

L364-367 you can’t use these sentences

L367-373 please don’t use leaps

L373-379 please discuss your results

L419 please remove it and its related phrases

The style of referencing does not match between the text and bibliographic sections. Through the text is author, year, but, the bibliographic is numbered. Also, please ensure that all referenced are included in the bibliographic section.

**Do you want your identity to be public for this peer review?** For information about this choice, including consent withdrawal, please see our Privacy Policy

Reviewer #1: No

Reviewer #2: No

---

## [Author Response · Author response to Decision Letter 1]

7 Aug 2025

Responses to reviewer comments

Reviewer #1:

1-The age distribution of participants in the abstract and the text of the article is different and not the same.

Answer – It was our mistake and has been corrected.

2- I suggest that a schematic image or time line of the study protocol and the jumps be included in the text of the article.

Answer – Thank you for your suggestion, it has been taken into account and the research procedure diagram has been included in the manuscript.

3. There is no discussion anywhere about the shoes of the participants. Are the shoes different in athletic and firefighter uniforms?

Answer – Thank you very much for this comment. Of course, this wasn't clearly stated. This has been corrected and supplemented in the research method.

4. Did anyone drop out of your study? What are the exclusion criteria?

Answer – No one drop out from the study, but six individuals were not qualified for the main study. Exclusion criteria were included in the research procedure diagram Fig 1 in manuscript.

Reviewer #2:

The authors have assessed three age groups pf firefighters with no support for age groups or age effect in the introduction. I can’t understand why three age groups of firefighters are included. Please clarify.

Answer – Occupational demands in firefighting persist across a wide age span, from younger recruits to experienced older personnel. Age-related changes in neuromuscular control, balance, and muscular strength may alter the way external loads are managed, potentially increasing susceptibility to overload. Despite this, limited research has examined how age interacts with physical demands typical of firefighting tasks, such as impact landings under external load.

Abstract

The structure of abstract does not follow a standard manner. This part must be rewritten.

Background, it is not logical that you completed this part with a question. Better to provide a background regarding to firefighters and their fire protection equipment which leads to overload and then your question. Results, please provide your findings objectively with p-values.

Answer – The abstract was revised in accordance with the reviewer's suggestions and the journal's guidelines.

Introduction

The flow of idea needs to be strengthened. After reading this section for several times, I cant understand what is the novelty and the rationale. This is simple to say that the authors have investigated the effect of two levels of clothes and three age groups on GRF. Please provide support for your outcomes. Moreover, I did not find any support for knee joint stability (H:Q) through the introduction. In this case, I could think the authors have collected any available variable with no rationale. Please rewrite the introduction accordingly.

Answer – Thank you for your suggestion. In the introduction part, explanations and rationale of the novelty were added.

Methods

Oreder of conditions is not randomized. Please acknowledge it in the study limitations.

Answer – Yes, You're right, that the order of conditions is not randomized and it is limitations of our study. We took into account the fire stations designated by the main fire service office. Your comment has been acknowledge it in the study limitations.

L130-137 It must not be appeared here.

Answer – The analysis of anthropometric variables was transferred to the research results

L215 and height

Answer – It was added

Discussion

L229 is unknown

L303-308 remove

Answer – It has been removed

L318-326 please discuss about your results, not review the literaturę

Answer – Some suggested text has been removed and some has been moved.

L335-336 different types of evaluating H/Q are existing. Mostly are isokinetic assessments. Here, based on your values and existing literature, please judge whether the values are low or not.

The reference Mbada et al., 2021 is not in the references list.

The reference Tabor et al., 2021 is not valid for your claim.

L341 please add reference

L352 Vy et al is not in the references list

Answer – I agree that we cannot base our theories on our own beliefs and our other research. Therefore, I have supplemented these theories with research from other authors. The reference Mbada et al. 2021 is in 25th position references list. There was a typo in the name. It has been corrected. The reference Vu et al. 2021 is in 10th position references list.

L360-361 you have not plantar flexion data. Please avoid leaps

L364-367 you can’t use these sentences

L367-373 please don’t use leaps

L373-379 please discuss your results

L419 please remove it and its related phrases

Answer – We agree with the reviewer's suggestion. The highlighted passages have been removed from the article.

The style of referencing does not match between the text and bibliographic sections. Through the text is author, year, but, the bibliographic is numbered. Also, please ensure that all referenced are included in the bibliographic section.

Answer – The references list has been supplemented and corrected.

---

## [Decision Letter · Decision Letter 1]

16 Sep 2025

Dear Dr. Iwańska,

Thank you for submitting your manuscript to PLOS ONE. After careful consideration, we feel that it has merit but does not fully meet PLOS ONE’s publication criteria as it currently stands. Therefore, we invite you to submit a revised version of the manuscript that addresses the points raised during the review process.

**ACADEMIC EDITOR:**

We look forward to receiving your revised manuscript.

Kind regards,

Shahnawaz Anwer, PhD

Academic Editor

PLOS ONE

Journal Requirements:

Additional Editor Comments:

Reviewer #1:

1- The age groups of the people selected in the main text of the article and the abstract are not the same.

2- What is the criteria for selecting people in the age category?

3- Nothing is said about the gender of the people.

4- Detailed study inclusion and exclusion criteria should be stated.

5- timeline and consort diagram of study must be present

6- Did the participants have musculoskeletal problems before the study? If so, what instrument was used and how was it measured?

Reviewer #2:

Thanks for your amendments。

The manuscript has been improved。 However, there are a few minor concerns listed below。

In the abstract

Move Objective after the background。 Include exact p-values instead of <0.01.

In the introduction

You can use (https://doi.org/10.1016/j.jbmt.2023.06.003) for injury or work related musculoskeletal disorder incidence.

Reference is needed for “The hamstring-to-quadriceps strength ratio (H:Q) is considered an important parameter in this context, as it affects knee joint stability and the ability to absorb vertical forces during landing. However, previous studies have not clearly linked H:Q ratios with GRF changes in firefighters under different external load”. Suggested reference: https://doi.org/10.1016/j.knee.2022.05.007

Methods

Move “A limitation of our study was the lack of randomization in the recruitment of study

participants” to the end of the discussion.

Reviewers' comments:

Reviewer's Responses to Questions

**Comments to the Author**

Reviewer #1: (No Response)

Reviewer #2: (No Response)

2. Is the manuscript technically sound, and do the data support the conclusions?

Reviewer #1: (No Response)

Reviewer #2: Yes

3. Has the statistical analysis been performed appropriately and rigorously?

Reviewer #1: (No Response)

Reviewer #2: Yes

4. Have the authors made all data underlying the findings in their manuscript fully available?

Reviewer #1: (No Response)

Reviewer #2: Yes

5. Is the manuscript presented in an intelligible fashion and written in standard English?

Reviewer #1: Yes

Reviewer #2: Yes

Reviewer #1: (No Response)

Reviewer #2: Thanks for your amendments.

The manuscript has been improved. However, there are a few minor concerns listed below.

In the abstract

Move Objective after the background. Include exact p-values instead of <0.01.

In the introduction

You can use (https://doi.org/10.1016/j.jbmt.2023.06.003) for injury or work related musculoskeletal disorder incidence.

Reference is needed for “The hamstring-to-quadriceps strength ratio (H:Q) is considered an important parameter in this context, as it affects knee joint stability and the ability to absorb vertical forces during landing. However, previous studies have not clearly linked H:Q ratios with GRF changes in firefighters under different external load”. Suggested reference: https://doi.org/10.1016/j.knee.2022.05.007

Methods

Move “A limitation of our study was the lack of randomization in the recruitment of study

participants” to the end of the discussion.

**Do you want your identity to be public for this peer review?** For information about this choice, including consent withdrawal, please see our Privacy Policy

Reviewer #1: No

Reviewer #2: No

---

## [Author Response · Author response to Decision Letter 2]

9 Nov 2025

Reviewer #1:

1- The age groups of the people selected in the main text of the article and the abstract are not the same.

2- What is the criteria for selecting people in the age category?

3- Nothing is said about the gender of the people.

4- Detailed study inclusion and exclusion criteria should be stated.

5- timeline and consort diagram of study must be present

6- Did the participants have musculoskeletal problems before the study? If so, what instrument was used and how was it measured?

Responses #1:

The age groups of the people selected in the main text of the article and the abstract was corrected. L 52-53; 171-173.

All criteria for selecting people in a given age category and detailed inclusion and exclusion criteria from the study were included in the main text.Thank you for your suggestion, it has been taken into account and the research procedure diagram has been included in the manuscript. L 153-163; 173-174.

The timeline and diagram of the study have been changed and placed in Figure 1. I hope that in its current form it will be clear and will provide an orderly overview of the course of our research.

Six people did not qualify for the main study due to exclusion by the medical doctor. This was due to cardiac problems and problems related to chronic diseases (diabetes).

Reviewer #2:

1 In the abstract

Move Objective after the background. Include exact p-values instead of <0.01.

2 In the introduction

You can use (https://doi.org/10.1016/j.jbmt.2023.06.003) for injury or work related musculoskeletal disorder incidence.

Reference is needed for “The hamstring-to-quadriceps strength ratio (H:Q) is considered an important parameter in this context, as it affects knee joint stability and the ability to absorb vertical forces during landing. However, previous studies have not clearly linked H:Q ratios with GRF changes in firefighters under different external load”. Suggested reference: https://doi.org/10.1016/j.knee.2022.05.007

3 Methods

Move “A limitation of our study was the lack of randomization in the recruitment of study

participants” to the end of the discussion.

Responses #2:

Thank you very much for your reviews and all your comments regarding our article. All suggestions have been incorporated into the main text. Significance levels in the abstract and in the main text were given as obtained during the analysis.

Thank you for sending new research in my area of interest. It was used it in manuscript and included it in the references.

---

## [Editor Report · Decision Letter 2]

25 Nov 2025

Overload of the lower limbs of firefighters as a result of external conditions

PONE-D-25-13106R2

Dear Dr. Iwańska,

We’re pleased to inform you that your manuscript has been judged scientifically suitable for publication and will be formally accepted for publication once it meets all outstanding technical requirements.

Kind regards,

Shahnawaz Anwer, PhD

Academic Editor

PLOS ONE
---

## [Editor Report · Acceptance letter]

PONE-D-25-13106R2

PLOS One

Dear Dr. Iwańska,

I'm pleased to inform you that your manuscript has been deemed suitable for publication in PLOS One. Congratulations! Your manuscript is now being handed over to our production team.

Kind regards,

on behalf of

Dr. Shahnawaz Anwer

Academic Editor

PLOS One